# Clinical Evidence for the Importance of the Wild-Type *PRPF31* Allele in the Phenotypic Expression of RP11

**DOI:** 10.3390/genes12060915

**Published:** 2021-06-14

**Authors:** Danial Roshandel, Jennifer A. Thompson, Rachael C. Heath Jeffery, Dan Zhang, Tina M. Lamey, Terri L. McLaren, John N. De Roach, Samuel McLenachan, David A. Mackey, Fred K. Chen

**Affiliations:** 1Centre for Ophthalmology and Visual Science, The University of Western Australia, Perth, WA 6009, Australia; danialroshandel@lei.org.au (D.R.); rachaelheathjeffery@lei.org.au (R.C.H.J.); tina.lamey@health.wa.gov.au (T.M.L.); terri.mclaren@health.wa.gov.au (T.L.M.); john.deroach@health.wa.gov.au (J.N.D.R.); samuelmclenachan@lei.org.au (S.M.); davidmackey@lei.org.au (D.A.M.); 2Ocular Tissue Engineering Laboratory, Lions Eye Institute, Nedlands, WA 6009, Australia; dana@lei.org.au; 3Australian Inherited Retinal Disease Registry and DNA Bank, Department of Medical Technology and Physics, Sir Charles Gairdner Hospital, Nedlands, WA 6009, Australia; jennifer.thompson3@health.wa.gov.au; 4Department of Ophthalmology, Royal Perth Hospital, Perth, WA 6000, Australia; 5Department of Ophthalmology, Perth Children’s Hospital, Nedlands, WA 6009, Australia

**Keywords:** retinitis pigmentosa, *PRPF31*, RP11, natural history, multimodal imaging, microperimetry, genotype-phenotype correlation

## Abstract

*PRPF31*-associated retinopathy (RP11) is a common form of autosomal dominant retinitis pigmentosa (adRP) that exhibits wide variation in phenotype ranging from non-penetrance to early-onset RP. Herein, we report inter-familial and intra-familial variation in the natural history of RP11 using multimodal imaging and microperimetry. Patients were recruited prospectively. The age of symptom onset, best-corrected visual acuity, microperimetry mean sensitivity (MS), residual ellipsoid zone span and hyperautofluorescent ring area were recorded. Genotyping was performed using targeted next-generation and Sanger sequencing and copy number variant analysis. *PRPF31* mutations were found in 14 individuals from seven unrelated families. Four disease patterns were observed: (A) childhood onset with rapid progression (*N* = 4), (B) adult-onset with rapid progression (*N* = 4), (C) adult-onset with slow progression (*N* = 4) and (D) non-penetrance (*N* = 2). Four different patterns were observed in a family harbouring c.267del; patterns B, C and D were observed in a family with c.772_773delins16 and patterns A, B and C were observed in 3 unrelated individuals with large deletions. Our findings suggest that the RP11 phenotype may be related to the wild-type *PRPF31* allele rather than the type of mutation. Further studies that correlate in vitro wild-type *PRPF31* allele expression level with the disease patterns are required to investigate this association.

## 1. Introduction

Autosomal dominant retinitis pigmentosa (adRP) with incomplete penetrance or variable phenotypic expression was first described in the late 1960s [1] and has since been linked to genetic variations at multiple genomic loci. The sixth locus to be associated with adRP was found on chromosome 19q [2,3,4]. This locus was mapped to 19q13.4 (RP11) and found to contain the human homologue of the yeast pre-mRNA splicing gene, Pre-mRNA Processing Factor 31 (OMIM: #606419, *prp31*, known as *PRPF31*) [5]. Mutations in the *PRPF31* gene account for up to 10% of adRP [6,7]. Designated RP11 (OMIM: #600138), this rod-cone dystrophy (RCD) usually presents in the teenage years but the age of onset may vary widely between and within families [8,9]. In addition, non-penetrance has been reported in many RP11 families [10,11,12]. It is widely accepted that haploinsufficiency is the underlying mechanism of retinal degeneration caused by the *PRPF31* mutation where expressivity of the normal allele determines the phenotype [13,14,15]. Augmentation of *PRPF31* expression by the normal allele in *PRPF31*^+/−^ induced pluripotent stem cell (iPSC)-derived retinal pigment epithelium (RPE) was shown to restore structure and function of the RPE in an in vitro study [16]. However, significant inter-individual variation in the natural history of RP11 may pose significant challenges in the design of future *PRPF31* therapeutic trials.

In addition to variant haploinsufficiency, an underlying mechanism in approximately 90% of RP11 families [15], a recent study showed that the *PRPF31* p.A216P missense mutation can lead to aggregation of wild-type and truncated protein in a mouse model [17], demonstrating a dominant-negative mechanism for this particular genotype. Moreover, Wheway et al. reported a significant correlation between the age of symptom onset and the type of *PRPF31* mutation and postulated that a dominant-negative mechanism and/or hypomorphic allele may lead to late-onset disease [18]. In the latter study, age of symptom onset was 8–12 years, 20–24 years and around 27 years in patients with nonsense/frameshift/indel variants, large deletions/splice variants and in-frame duplications/insertions/missense variants, respectively. Thus the authors concluded that the type of *PRPF31* mutation predicts phenotypic severity highlighting the importance of a non-haploinsufficiency mechanism [18].

Previous studies have shown an exponential decline in Goldmann residual visual field area, full-field electroretinography (ERG) cone amplitude and spectral-domain optical coherence tomography (SD-OCT) ellipsoid zone (EZ) area [19,20]. However there were no reports on individualized progression rates or cross-sectional inter-individual variations in patients of a similar age [19] or disease duration [20]. The natural history of RP11 can be highly variable among patients of the same family regardless of their age of onset and disease duration [21]. We previously found the span of the residual EZ and area of the hyperautofluorescent ring (HAR) to be potential structural endpoints for future clinical trials [21]. However, that report was limited to a single pedigree.

Herein, we report multimodal imaging and microperimetry findings in seven unrelated RP11 families in order to explore the genotype-phenotype correlation and natural disease progression.

## 2. Materials and Methods

### 2.1. Participants

Patients were recruited from the Western Australian Retinal Degeneration (WARD) study, a prospective observational cohort study. Existing imaging data prior to enrolment were also collected for analysis. Diagnosis of RP was made by an inherited retinal disease specialist (FKC) based on history, retinal findings and ERG. Genetic diagnosis was established through the Australian Inherited Retinal Disease Registry and DNA Bank (AIRDR). The study protocol was approved by the Institutional Review Board of the University of Western Australia (RA/4/1/7916, 2021/ET000151) and Sir Charles Gairdner Hospital Human Research Ethics Committee (approval number 2001-053). Written informed consent was obtained from all participants and the study adhered to principles of the declaration of Helsinki.

### 2.2. Clinical Assessment

Functional and structural assessments were performed using a protocol previously described [21]. Best-corrected visual acuity (BCVA) was measured using the Early Treatment Diabetic Retinopathy Study (ETDRS) chart at 4 m. Snellen equivalents were reported where appropriate. For patients who were unable to read the ETDRS chart, their ability to count fingers (CF), perceive hand motions (HM) or perception of light (PL) were recorded. Standard automated perimetry (24–2 pattern, 54 test loci, III-white stimulus, stimulus intensity range 0–40 dB) was performed using the Humphrey Field Analyzer (HFA-II 750, Carl Zeiss Meditec GmbH, Jena, Germany). HFA 24–2 mean deviation (MD), which is calculated based on the normative values provided on the device, was recorded. Baseline and follow-up microperimetry (Macular Integrity Assessment, MAIA, Centervue, Padova, Italy) with a stimulus intensity range of 0–35dB were performed. The large 10–2 (68 test loci) and the small radial (37 test loci, 37R) grid patterns were used to map the retinal sensitivity profile within the macular (central 20° field) and foveal (central 6° field) regions, respectively (Appendix A). Retinal sensitivity was defined as normal (≥ 26 dB), scotoma (0–25 dB) or dense scotoma (<0 dB), according to the manufacturers’ recommendation. The average retinal sensitivity of the test grid (mean sensitivity; MS) and the number of seeing loci defined as threshold ≥ 0 dB were recorded. Full-field ERG (RETIport 3.2, Roland Consult, Brandenburg, Germany or in-house custom built) was recorded in accordance with International Society for Clinical Electrophysiology of Vision (ISCEV) standards [22].

Serial ultra widefield (UWF) color and autofluorescence imaging (P200Tx and California, Optos plc, Dunfermline, UK) were performed to document the severity of intraretinal pigmentation and loss of normal fundus autofluorescence. SD-OCT (Spectralis OCT2, Heidelberg Engineering, Heidelberg, Germany) was performed on all patients using a horizontal line scan (100 frames averaged) and a raster volume scan covering the central 30° × 25° area (9 frames average per line scan, 61 lines per cube, 130 µm separation). Nasal and temporal ellipsoid zone (EZ) limits were determined manually on the foveal-centered horizontal line scans for EZ span measurement. Short-wavelength (excitation λ = 488 nm) and near-infrared (excitation λ = 787 nm) fundus autofluorescence (AF, HRA2, Heidelberg Engineering) were performed in all patients, capturing the central 30° and 55° of retina. Both short-wavelength AF (SWAF) and near-infrared AF (NIAF) images were examined for the presence of a hyperautofluorescent ring (HAR) and the outer boundaries of this HAR were manually delineated. The area of the HAR was measured with in-built software (HEYEX v1.9.14.0, Heidelberg Engineering). 

### 2.3. DNA Analysis and Pathogenicity Assessment

Genomic DNA samples were collected, processed and stored as previously described [23]. Genomic DNA was analysed using disease-specific next generation sequencing panels [24] covering all exons and flanking intronic regions and relevant deep-intronic regions of target genes, and copy number variant analysis or Array CGH, as detailed in Appendix A. Identified candidate mutations were confirmed in probands and family members by Sanger sequencing or TaqMan qPCR. Genetic testing was performed by Molecular Vision Laboratory (Hillsboro, OR, USA), Casey Eye Institute (Portland, OR. USA), or the Australian Genome Research Facility (Perth, Australia). Sequencing results were aligned to the reference sequence NM_015629.3, with nucleotide position 1 corresponding to the A of the ATG translation initiation codon. Variant nomenclature is reported in accordance with the recommendations of the Human Genome Variation Society [25] and pathogenicity was assessed as described previously [26] and interpreted according to the joint guidelines of the American College of Medical Genetics and Genomics and the Association for Molecular Pathology (ACMG/AMP) [27] and associated literature [28,29,30,31].

### 2.4. Statistical Analysis

Data were recorded in Statistical Package for the Social Sciences (SPSS) version 23 (SPSS/IBM, Inc., Chicago, IL, USA). Data distribution was examined visually for normality and student’s t-test was used to calculate mean (SD) of quantitative parameters. Annual change of each endpoint was calculated using linear regression analysis and absolute (β estimate) and relative (percent of baseline) values were reported. Only the right eye progression data were presented, unless otherwise specified.

## 3. Results

A total of 14 mutation carriers from seven unrelated RP11 families were enrolled in this study (Figure 1). The spectrum of mutations and phenotypes were summarized in Table 1. Two unrelated cases (1651, 1816) did not show any symptoms and/or signs of RP at their last examination and were subsequently labelled as non-penetrant carriers (NPCs) (Appendix A).

### 3.1. DNA Analysis and Pathogenicity Assessment

Genetic testing revealed five previously reported mutations and two novel variants (Appendix A). A nonsense-mediated mRNA decay (NMD) mechanism was predicted for the previously reported c.267del (p.(Glu89Aspfs*11)) [32] and c.772_773delins16 (p.(Thr258Glnfs*68)) [33] mutations. The splice site variants (c. -9+1G>T and c.527+1G>T) have been shown to result in production of truncated proteins [34,35,36]. We also detected large deletions in three families, including two novel variants that were classified as variants of uncertain significance (VUS) according to the ACMG/AMP criteria. The previously reported (Exon 2–3 deletion [37,38]) and one of the novel (Exon 2–8 deletion) variants would remove the start codon (start-loss variants), hence they are expected to cause no-go mRNA decay. Similarly, the third large deletion (Exon 4–19 deletion) excludes the last six exons including the stop codon (stop-loss variant), and would be expected to cause non-stop mRNA decay [39].

### 3.2. Baseline Clinical Features

#### 3.2.1. Small Deletion/Deletion-Insertion

A small deletion or deletion-insertion was detected in two families. The proband of the family with the c.267del mutation (Figure 1A) showed advanced RP with severe macular involvement and a BCVA of CF at the time of baseline imaging (Appendix A). She had a history of nyctalopia from 6 years of age, progressing towards severe peripheral vision loss and profound visual impairment by the age of 60. The proband’s daughter also developed symptoms of RP at 6 years of age. Baseline imaging showed widespread pigmentation and loss of AF, a small central hyper AF island and widespread atrophy of the RPE and outer retina in both eyes (Figure 2C). The Humphrey visual field (HVF) test showed peripheral visual field loss in both eyes (Appendix A). Foveal MS was 4.7 dB and 9.4 dB in the right and left eyes, respectively (Appendix A). The proband’s son was diagnosed with RP at age 49. Extensive loss of normal AF with a small central hyper AF island was noted on NIAF imaging in both eyes, whilst SD-OCT showed intraretinal fluid and severe EZ atrophy (EZ span = 1505 µm) in the right eye and intraretinal fluid with a residual EZ line spanning 2529 µm in the left eye (Figure 2D). HVF demonstrated bilateral visual field constriction with a small central island of vision (Appendix A). His two daughters carried the mutation and showed typical signs and symptoms of RP in the third decade. Whilst the older sister had a large HAR and EZ line with normal foveal structure (Figure 2B), the younger sister had a small area of central hyper AF, severe EZ atrophy and cystoid macular oedema (CMO) in both eyes (Figure 2A). Microperimetry of the older sister showed peripheral macular sensitivity loss (MS 19.1 dB and 17.8 dB in the right and left eyes, respectively) (Appendix A) and normal foveal sensitivity (MS 29.1 and 28.2 in the right and left eyes, respectively) (Appendix A). The proband’s nephew carried the same mutation, but had no signs or symptoms of RP at age 41 (Appendix A).

Variant c.772_773delins16 was found in one family (Figure 1B) including two affected siblings and their NPC father (Appendix A). The older sibling was asymptomatic and showed normal foveal and macular MS (Figure 3L,M), large HAR only detectable on UWF imaging (Figure 3O–Q) and loss of the peripheral EZ on widefield SD-OCT (Figure 3R,S), whilst the younger sibling demonstrated reduced macular MS (Figure 3C), constricted macular HAR (Figure 3F–H) and shortened EZ in both eyes (Figure 3I). Both siblings had features of RCD on full-field ERG (Figure 3A,J).

#### 3.2.2. Splice Site Mutations

A splice site mutation was found in two unrelated families (Figure 1C,D). One patient with c.-9+1G>T mutation had a history of nyctalopia from the age of 18 and had undergone a right penetrating keratoplasty (PK) for the management of advanced keratoconus at 26 years. SD-OCT showed a large (4500 µm) residual EZ line in both eyes at 61 years. In addition, a large macular HAR and preserved central visual field were observed in the left eye (Figure 4A). The second patient carried the c.527+1G>T mutation and was diagnosed with RP at 3 years. Baseline multimodal imaging at 70 years revealed end-stage RP with myopic features in both eyes (Figure 4B).

#### 3.2.3. Large Deletions

Three large deletions were identified in three unrelated families (Figure 1E–G). Patient 1705 presented with a history of nyctalopia from 26 years of age. Multimodal imaging at 62 years revealed widespread retinal atrophy with a small central island of preserved outer retinal structures and partially preserved foveal function in both eyes (Figure 4C). Genetic testing revealed exon 2–3 deletion. Patient 1164 reported peripheral visual field loss from 35 years. Baseline evaluations demonstrated a well-preserved BCVA and foveal sensitivity on 37R MAIA (right and left MS = 23.3 dB and 23.5 dB, respectively) and a dense perifoveal scotoma on 10–2 MAIA at 61 years. The residual EZ span and NIAF HAR area in the right eye were 3553 µm and 9.5 mm^2^, respectively (Figure 4D). A heterozygous deletion of exons 2–8 was detected in this patient. Patient 1175 was diagnosed with RP at 4 years and presented to our clinic with severely diminished macular function at 35 years. A small EZ line (1014 µm) and HAR (2.1 mm^2^) were detected in the right eye (Figure 4E). Genetic analysis revealed heterozygous deletion of exons 9–14. Appendix A summarizes baseline HVF 24–2 MD, macular and foveal MS and number of seeing loci. SD-OCT and SLO AF imaging were performed in 12 and 9 patients, respectively. EZ line was detected in 9/12 patients (16 eyes), while HAR was detected in 6/9 patients (10 eyes) using NIAF and 5/9 patients (eight eyes) using SWAF (Appendix A).

### 3.3. Natural History of Disease Progression

Serial macular and foveal MS, residual EZ span and NIAF HAR area measurements were available in a number of patients with different mutations (Appendix A). Exceptions included patient 1473, who had poor image quality in the right eye due to irregular astigmatism post-PK and patient 1705 who had no detectable EZ line in the right eye at baseline imaging. A statistically significant decline in foveal MS (0.4 dB/year decline over 4.8 years) was observed only in the patient with the exon 9–14 deletion (patient 1175). Both patients with c.772_773delins16 showed more than 2 dB/year increase in foveal and macular MS, though the follow-up period was equal to or less than 12 months (Appendix A and Figure 5).

EZ span and HAR area showed statistically significant decline with age in all patients with more than 2 years of follow up. The range of absolute (or relative) rate of decline in the EZ span and HAR area was 65–174 µm (or 2.0–6.4%) and 0.2–0.6 mm^2^ (or 2.1–9.5%), respectively. Of interest, patient 1164 with exon 2–8 deletion showed a lower EZ span and HAR area decline as compared to patient 1175 with exon 9–14 deletion (Appendix A and Figure 6). Figure 7 depicts the baseline and longitudinal foveal and macular MS, EZ span and HAR area plotted against age for the whole cohort.

### 3.4. Phenotype Patterns

Based on the age of symptom onset and multimodal imaging findings, four patterns of disease were identified including: (A) childhood onset with rapid progression (*N* = 4), (B) adult-onset with rapid progression (*N* = 4), (C) adult-onset with slow progression (*N* = 4), and (D) non-penetrance (*N* = 2). Patients in group A had no or barely detectable EZ with a BCVA of 20/50 or worse and MS less than 2 dB by 50 years (patients 1150 and 1175) or CF by 80 years (patients 1313 and 1708). Patients in group B presented with a reduced EZ span of 4000 µm by age 17 (patient 1681), 3000 µm by age 29 (patient 1332), 2500 µm by age 60 (patient 1477) and 750 µm by age 70 (patient 1708). Group C (*N* = 4) revealed either a mild asymptomatic phenotype (normal MS and large EZ and HAR) at age 18 (patient 1757) or a residual EZ span greater than 3500 µm and MS 8–19 dB by age 34–63 (patients 1506, 1473 and 1164). All phenotypes were observed in our family with the c.267del variant, phenotypes B, C and D were observed with the c.772_773delins16 variant and phenotypes A, B and C were observed in three unrelated patients with large deletions. Table 2 summarizes the number of patients allocated to each phenotype pattern.

## 4. Discussion

This study characterises RP11 patterns of disease using multimodal retinal imaging in seven unrelated adRP families with different classes of *PRPF31* mutations and suggests that RP11 phenotype can be classified into early- or late-onset and slow or rapid progression. Such classification may be useful in future genotype-phenotype correlation and natural history studies, as well as RP11 clinical trials. We demonstrated variations in phenotypic expression and the natural history of the disease both within and between families, irrespective of the type of mutation. This phenotypic variability observed suggests variations in expression from the unaffected allele, rather than the mutated allele, influences the clinical presentation and severity of RP11.

Although pathogenicity of the novel large deletions is not assured, it is likely that they result in start-loss and stop-loss variants which would be expected to undergo no-go or non-stop mRNA decay. However, due to the possibility of alternative splicing, we cannot rule out production of alternative transcripts. Further in vitro studies are required to explore the exact effect of these multi-exon deletions at the level of mRNA and protein, as well as PRPF31 function.

### 4.1. RP11 Phenotype Varies in Patients with Identical or Similar Mutations

A bimodal phenotype that included both asymptomatic (non-penetrant) and symptomatic RP11 patients was initially reported [3]. Further studies later revealed a spectrum of age of onset, BCVA and residual peripheral visual field in symptomatic patients [8,40,41,42,43]. This age range was reported in cohorts with different *PRPF31* mutations [20,44] as well as within members of the same family [45]. In the present study, age of symptom onset was 3–49 years across the cohort, 6–49 years in patients from one family carrying the c.267del mutation and 4–35 years in patients with large deletions, showing a remarkable variation both between and within families.

Using multimodal imaging, we observed variations in 2 families with more than one affected member (Figure 2 and Figure 3). Interestingly, patient 1757 showed signs of RP whilst remaining asymptomatic. The EZ atrophy and HAR were best visualized using UWF FAF and OCT (Figure 3). This finding emphasizes the utility of widefield imaging for the diagnosis of early-stage disease in asymptomatic family members. In addition, 3 unrelated patients with large deletions showed significant differences in their functional and structural findings (Figure 4). Kiser et al. reported an exponential age-dependent decline in EZ area in a cohort of 15 patients, eight of whom had serial measurements, with different *PRPF31* mutations. Although they did not report individual data for each patient, scatterplots revealed a large variation in EZ area between patients around 50 and 60 years old [20]. We have previously reported variations in the EZ span in 12 patients from a single large family and found remarkable differences between patients of a similar age [21], which is in agreement with our present study. However, variation in multimodal imaging endpoints in RP11 families with different mutations was not investigated previously.

### 4.2. RP11 Progression May Not Follow a First-Order Exponential Curve

Kiser et al. proposed a similar course of disease progression once the degenerative process begins at a “critical age”, probably secondary to an environmental trigger [20]. This “one-hit” model of cell death in neuronal degeneration was proposed by Clarke et al. based on different retinal degeneration mouse models [46]. According to this theory, the rate of cell death follows an exponential curve rather than a sigmoid curve. However, the relatively stable retinal structure and function during the 12 months’ follow up in patient 1757, despite rapid disease progression in his sibling of a similar age, contradicts this “one-hit” model for RP11. In addition, we reported a rapid disease progression in 3 60-year-old patients (patients 1473, 1705 and 1164) decades after their disease onset, providing further evidence against this model. However, patient-reported age of onset may not be a reliable indicator of actual onset and the duration of the degenerative process [20]. More likely, retinal damage in these patients progressed enough to produce symptoms at the reported age of onset. Hence, it is possible that the degenerating process slowed or even paused at some point or followed a sigmoidal curve rather than exponential. Longer follow up with larger cohorts is required to fully explore the disease course and validity of the “one-hit” model of RP11.

### 4.3. Implication of Wild-Type vs. Mutant Allele in RP11 Phenotype

Although approximately 90% of *PRPF31* mutations have been predicted to result in null alleles through a nonsense-mediated mRNA decay (NMD) mechanism [15], the penetrance of the retinal phenotype may depend on the expressivity of the wild-type allele [13,14,15]. Expression of *PRPF31* is continuous and highly variable in the general population, with a five-fold change between the lowest and the highest expression levels reported in lymphoblastic cell lines [47]. Hence, coinheritance of a mutant null allele in combination with a highly expressive functional *PRPF31* allele could result in an expression level that is adequate for retinal function, while coinheritance of a null allele and a functional allele with low expressivity could result in expression levels falling to below the critical threshold required for retinal maintenance, leading to photoreceptor degeneration [13,14,47]. Expression of the normal allele in asymptomatic (non-penetrant) carriers was up to two-fold higher than affected patients [13,14,15]. This may explain variations in penetrance amongst patients within the same family and highlights the role of the unaffected allele in determining the phenotype. Multiple *cis*- and *trans*-acting factors such as *CNOT3* [48,49] and minisatellite repeat (MSR) elements [50] regulate the expression of *PRPF31*. CNOT3 binds to *PRPF31* core promoter and downregulates its expression, whilst inhibition of *CNOT3* upregulates *PRPF31* expression [48,49]. These findings suggest that epistatic factors have a role in *PRPF31* expression by the wild-type allele and should be considered in future investigations. In our cohort, MSR1 genotyping in three sibling pairs from family 0255 (1477 and 1150; 1506 and 1332) and family 3200 (1757 and 1681) who had different phenotypic presentations showed a 3/3 genotype in all of them (unpublished data).

Additionally, Wheway et al. proposed that some *PRPF31* mutations may influence the age of diagnosis by exerting a dominant negative effect in conjunction with haploinsufficiency. For instance, patients with large-scale deletions have the latest age of diagnosis (due to haploinsufficiency). The authors postulated that in some cases an additional dominant negative mechanism may be involved, caused by a mutant protein either containing amino acid changes, or alternatively spliced, or prematurely truncated due to frameshift/nonsense mutations in the terminal exon or the C-terminal portion of the penultimate exon, resulting in earlier diagnosis [18]. To date, the only *PRPF31* mutation with a confirmed dominant negative effect is a missense variant reported in the *Prpf31^p^.^A216P/+^* mouse model [17]. Although all of the mutations observed in our study are predicted to cause loss of function, functional characterisation is still required to confirm these predictions. The intra-familial variation we observed may provide a clue as to the role of the wild-type allele, in determining the natural history of the disease.

### 4.4. Limitations

Uncertainty about the pathogenicity and functional consequences of the novel large multi-exon deletions is a potential limitation of our study. Pathogenicity assessment was hindered due to the novel nature of the variants and lack of identified deletion breakpoints, and their absence from databases of healthy individuals was not taken into account as population data for large deletions may not be detected by the sequencing methodology. In addition, we reported the clinical presentation of seven different mutations in 5 families; however, follow-up examinations were not always available in many of the cases. Another limitation of our study was the lack of patients with juvenile-onset disease between 7–16 years old. Hence, our suggested classification only applies to the studied sample and cannot be extrapolated to all RP11 patients. In addition, the follow-up duration was not long enough in most of our patients to show the natural course of the disease using different endpoints.

## 5. Conclusions

The RP11 phenotype and progression rate may vary not only in patients with different classes of mutations, but also in those with a similar mutation class or even the same mutation. Our results provide novel clinical evidence supporting a strong correlation between the wild-type allele rather than the mutated allele, determining the phenotype using multimodal imaging. The use of larger cohorts and long-term natural history data may help to define inclusion and exclusion criteria and the optimal window for therapeutic intervention based on disease onset and progression rates. In addition, gene expression and functional studies may provide additional information regarding the relationships between mutant and wild-type alleles and their role in the phenotypic expression of RP11. We recommend further investigation of tissue PRPF31 levels using patient-derived retinal tissue [51] that may provide additional clues towards a better understanding of these relationships and finding potential therapeutic targets.

## Figures and Tables

**Figure 1 genes-12-00915-f001:**
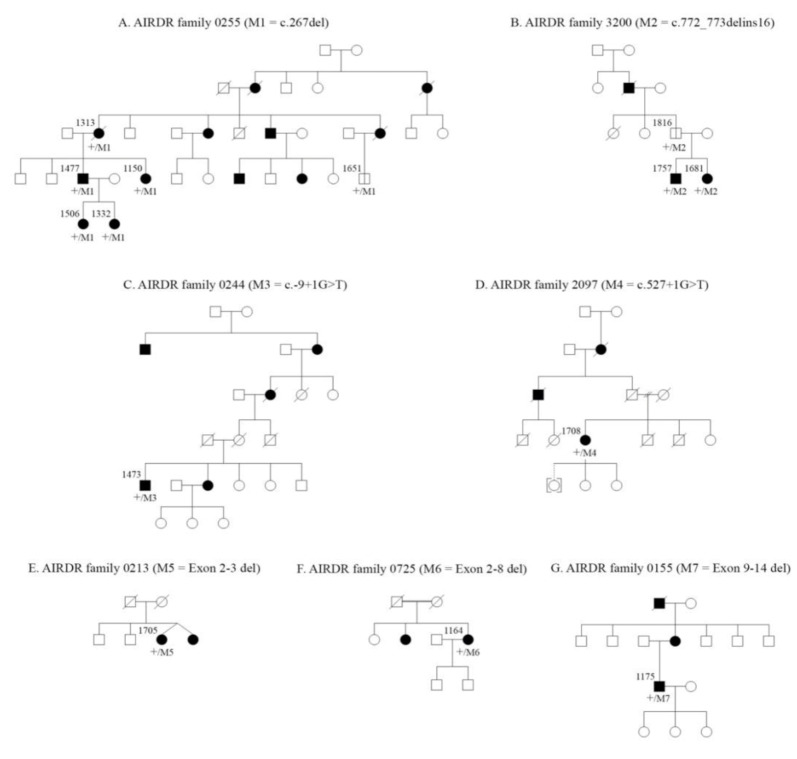
Pedigrees of the autosomal dominant retinitis pigmentosa families diagnosed with *PRPF31*-associated retinopathy. Patients enrolled in the present study are marked with WARD study ID.

**Figure 2 genes-12-00915-f002:**
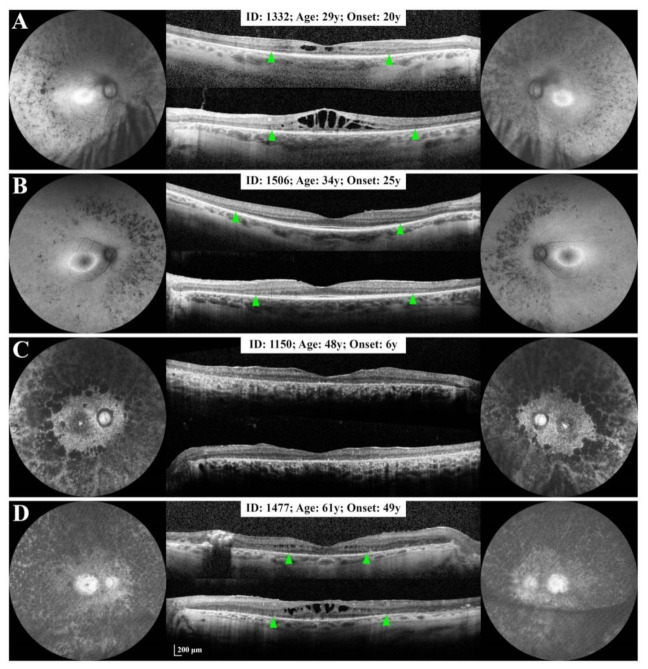
Ultra widefield (**A**,**D**) and 55° short-wavelength (**B**,**C**) autofluorescence imaging and spectral-domain optical coherence tomography of 4 affected members of family 0255. Panels (**A**,B) show 2 siblings with adult-onset disease. The younger sister (**A**) shows a more severe phenotype compared to the older sibling (**B**), despite similar duration of symptoms. The patient with childhood-onset disease (**C**) presented with complete disappearance of the ellipsoid zone (EZ) at age 48 years, whilst her sibling with adult-onset disease retained a greater than 2500 µm EZ span in the left eye at age 61 years (**D**). Green arrowheads show nasal and temporal EZ endings. Bilateral increased autofluorescence signal in the optic nerve heads in Panels (**C**,**D**) is due to the use of green excitation induced autofluorescence and brightness adjustment to improve contrast.

**Figure 3 genes-12-00915-f003:**
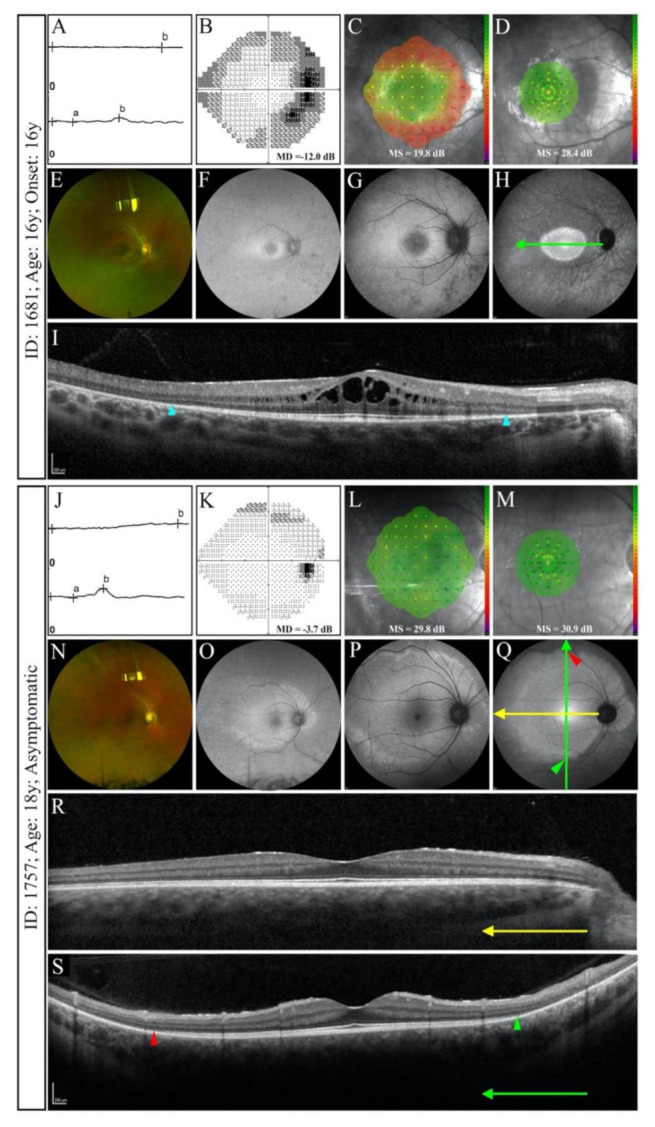
Functional and structural findings in two siblings with c.772_773delins16. The younger sibling presented with profoundly reduced rod response and minimal cone response in full-field electroretinography (ffERG; (**A**)), constricted visual field in 24-2 Humphrey test (**B**), reduced macular sensitivity in 10-2 MAIA (**C**) and normal foveal sensitivity in 37R MAIA (**D**). Bone spicule pigmentation with greater density at nasal side was noted on ultra widefield (UWF) colour photograph (**E**) and a typical macular hyperautofluorescence ring (HAR) was observed on UWF autofluorescence (AF; (**F**)), short-wavelength AF (SWAF; (**G**)) and near-infrared AF (NIAF; (**H**)). However, the borders of HAR could be delineated only on NIAF (H). Spectral-domain optical coherence tomography (SD-OCT) showed shortening of the ellipsoid zone (EZ; (**I**)). The asymptomatic older sibling showed similar ffERG findings (**J**), but Humphrey field (**K**) and microperimetry (**L**,**M**) indices were less severely affected. Although retinal pigmentation was not detected (**N**), a large HAR incorporating the optic disc was observed on UWF AF (**O**). Although the HAR was visible on both SWAF (**P**) and NIAF (**Q**), the temporal border fell beyond the 55° imaging field. EZ appeared normal on horizontal SD-OCT scan (**R**), however, borders of the EZ were detectable on vertical widefield SD-OCT (**S**). Arrows show the position and direction of the B-scans. Arrowheads indicate borders of preserved EZ.

**Figure 4 genes-12-00915-f004:**
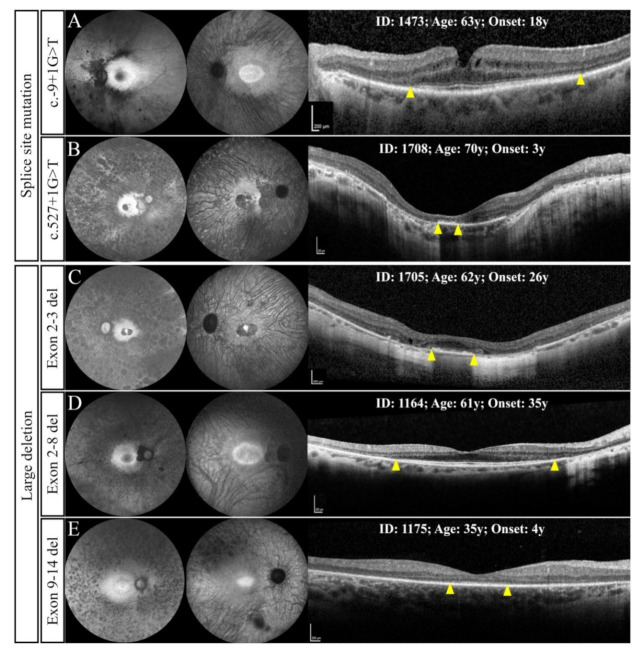
Ultra widefield autofluorescence (UWF AF) and spectral-domain optical coherence tomography findings in patients with splice site mutations (**A**,**B**) and large deletions (**C**–**E**). Whilst patient 1473 revealed a large hyperautofluorescent ring (HAR) and wide ellipsoid zone (EZ) span (**A**), patient 1708 showed extensive loss of normal autofluorescence and barely detectable EZ (B). Similarly, patients with large deletions revealed wide variation, including no HAR and small EZ (**C**) at age 62 years, large HAR and EZ at age 61 years (**D**) and small HAR and EZ at age 35 years (**E**). Yellow arrowheads show nasal and temporal EZ endings.

**Figure 5 genes-12-00915-f005:**
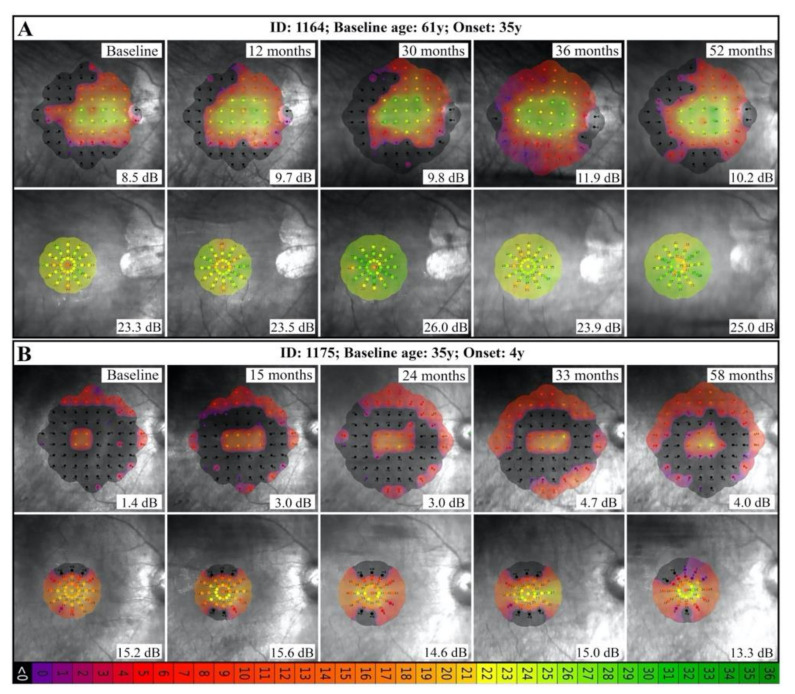
Serial 10-2 (macular) and 37R (foveal) MAIA in 2 patients with large deletions show short-term variation and long-term stability. Interestingly, the 61-year-old patient with adult-onset disease (**A**) had remarkably greater foveal and macular function compared to the 35-year-old patient with childhood-onset disease (**B**). Mean sensitivities are shown on bottom-right corner of each image.

**Figure 6 genes-12-00915-f006:**
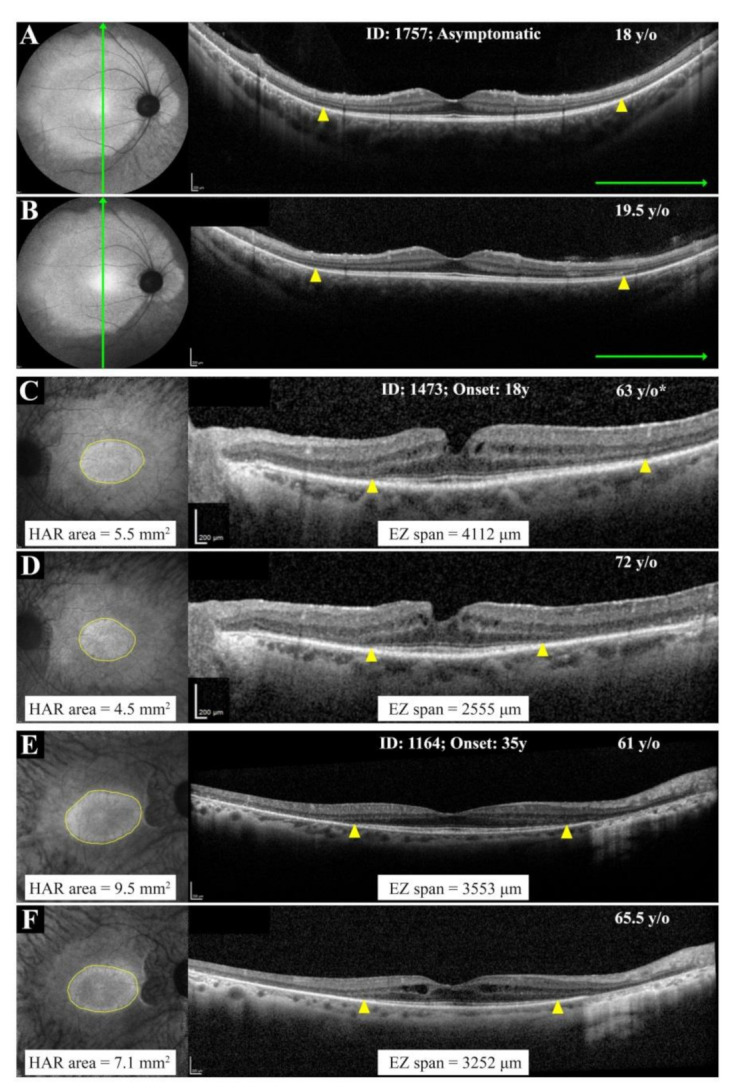
Structural disease progression using near-infrared autofluorescence (NIAF) imaging and spectral-domain optical coherence tomography (SD-OCT) in 3 patients with adult-onset disease. The asymptomatic patient showed stable hyperautofluorescent ring (HAR) and ellipsoid zone (EZ) extent over 1.5 years’ follow up (**A**,**B**). Since EZ endings were detectable only on vertical widefield SD-OCT and HAR temporal border fell beyond the imaging field, measurements were not reported. In contrast, patients 1473 (**C**,**D**) and 1164 (**E**,**F**), both with large deletions, showed variable rates of disease progression on both NIAF and SD-OCT. * The baseline NIAF imaging in patient 1473 was performed at age 69 years. Yellow arrowheads show nasal and temporal EZ endings.

**Figure 7 genes-12-00915-f007:**
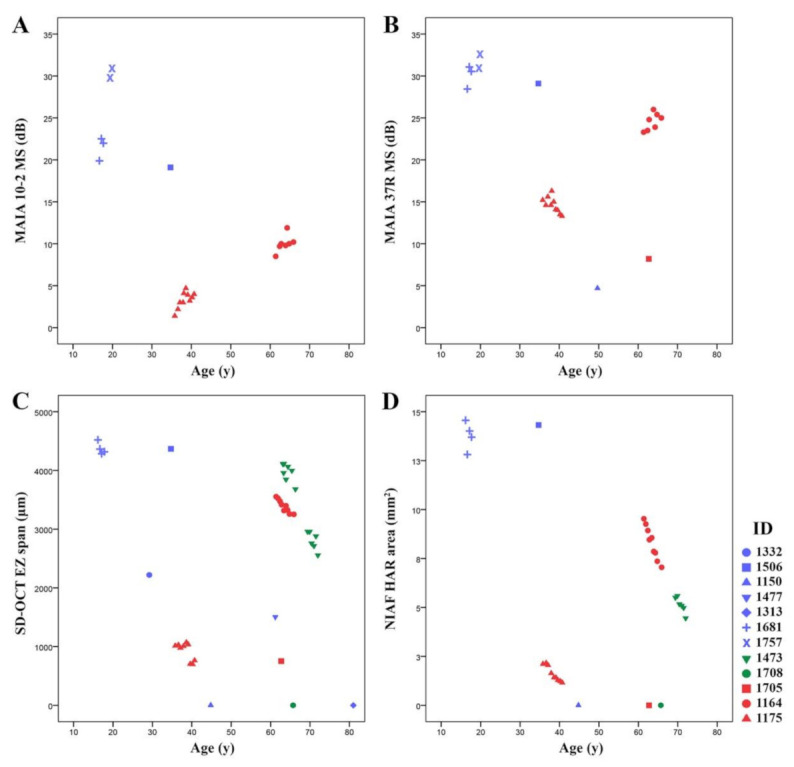
Scatterplots showing 10–2 (macular) and 37R (foveal) MAIA mean sensitivity (MS), spectral-domain optical coherence tomography (SD-OCT) ellipsoid zone (EZ) span and near-infrared autofluorescence (NIAF) hyperautofluorescent ring (HAR) area against age in all available patients, including serial measurement in some patients. Colours of markers indicate genotype groups including small deletion and deletion/insertion (Blue), splice site mutations (Green) and large deletions (Red). An overall decline of macular (**A**) and foveal (**B**) MS with increasing age was noted in the whole cohort. However, both methods showed short-term variations in individual patients. Both EZ span (**C**) and HAR area (**D**) showed age-dependent decline in those with available serial measurements. However, substantial variation was noted between patients and genotypes.

**Table 1 genes-12-00915-t001:** Demographics and baseline clinical characteristics of *PRPF31* mutation carriers including penetrant (*N* = 12) and non-penetrant (*N* = 2).

AIRDR Pedigree ID	WARD Study ID	Age (y) *	Sex	Onset (y) **	BCVA (ETDRS) ***		Lens ^§^		Mutation	Phenotype
					RE	LE	RE	LE		
0255	1332	29	F	20	68 (20/50)	68 (20/50)	PSCC +	PSCC +	c.267del	B
	1506	34	F	25	84 (20/20)	74 (20/32)	Clear	Clear	c.267del	C
	1651 ^†^	41	M	-	92 (20/16)	92 (20/16)	Clear	Clear	c.267del	D
	1150	48	F	6	45 (20/125)	62 (20/63)	IOL (44)	IOL (43)	c.267del	A
	1477	61	M	49	50 (20/100)	42 (20/160)	IOL (NA)	IOL (NA)	c.267del	B
	1313	81	F	6	CF	CF	IOL (56)	IOL (54)	c.267del	A
3200	1681	16	F	16	83 (20/25)	85 (20/20)	Clear	Clear	c.772_773delins16	B
	1757	18	M	-	84 (20/20)	83 (20/25)	Clear	Clear	c.772_773delins16	C
	1816 ^†^	56	M	-	NA	NA	Clear	Clear	c.772_773delins16	D
0244	1473	63	M	18	HM ^‡^	35 (20/200)	NSC +	NSC +	c.-9+1G>T	C
2097	1708	70	F	3	CF	CF	IOL (56)	IOL (56)	c.527+1G>T	A
0213	1705	62	F	26	35 (20/200)	64 (20/50)	IOL (56)	IOL (51)	Exon 2–3del	B
0725	1164	61	F	35	70 (20/40)	70 (20/40)	IOL (58)	IOL (58)	Exon 2–8del	C
0155	1175	35	M	4	59 (20/63)	64 (20/50)	PSCC +	PSCC +	Exon 9–14del	A

* Age at baseline imaging ** patient-reported age of onset of symptoms *** Snellen equivalents are shown in parentheses ^§^ Patients’ age (years) at the time of cataract extraction are shown in parentheses ^†^ Non-penetrant carrier ^‡^ History of penetrating keratoplasty due to keratoconus. BCVA = best-corrected visual acuity; CF = counting fingers; ETDRS = Early Treatment Diabetic Retinopathy Study; HM = hand motion; IOL = intraocular lens; LE = left eye; NA = not available; NSC = nuclear sclerosing cataract; PSCC = posterior subcapsular cataract; RE = right eye. A = early-onset with rapid progression; B = adult-onset with rapid progression; C = adult-onset with slow progression; D = Non-penetrant carrier.

**Table 2 genes-12-00915-t002:** Allocation of patients to phenotype categories.

Mutation Class	Phenotype A	Phenotype B	Phenotype C	Phenotype D	Total
Small deletion or deletion/insertion	2	3	2	2	9
Splice site mutations	1	0	1	0	2
Large deletions	1	1	1	0	3
Total	4	4	4	2	14

Phenotype A = early-onset with rapid progression; phenotype B = adult-onset with rapid progression; phenotype C = adult-onset with slow progression; phenotype D = Non-penetrant carrier.

## Data Availability

This study did not report any data.

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
