# Peer review of "Clinical Evidence for the Importance of the Wild-Type PRPF31 Allele in the Phenotypic Expression of RP11"

_genes, 2021, doi:10.3390/genes12060915_

Round 1
Reviewer 1 Report
This is a nice study that characterized RP11 patterns of disease using multimodal retinal imaging in 7 unrelated adRP families with different classes of PRPF31 mutations. It was well-written, but I have a few suggestions for improvement.
- Please give the stimulus intensity range of the MAIA and HFA.
- In the legend of Figure 1, it reads “….present study are marked with WARD study ID.” What is WARD? Please define in the manuscript.
- Please indicate how “normal” is assigned to perimetry results. Did you establish a normative database or did you use values provided on the device?
- Figure 2C and D have hyper AF optic disks. Is this significant to the phenotype? Is it ONH drusen? If this is a finding, please include in the manuscript to further expand the description of the phenotype associated with PRPF31.
- Figure 3, please describe and indicate A, B, C etc in the text for the given panels. Please tell what the arrows are indicating. The legend is confusing when 2 parenthesis are side-by-side. For example, “auto-fluorescence (AF) (F),” should be changed to “auto-fluorescence (AF; F)”. Or, put the (F) before discussing what is observed. So, “Bone spicule pigmentation with greater density at na-sal side was noted on ultra-widefield (UWF) colour photograph (E) and a typ-ical macular hyperautofluorescence ring (HAR) was observed on UWF auto-fluorescence (AF) (F), short-wavelength AF (SWAF) (G) and near-infrared AF (NIAF) (H). However, the borders of HAR could be delineated only on NIAF (H). Spectral-domain optical coherence tomography (SD-OCT) showed short-ening of the ellipsoid zone (EZ) (I).”
would be changed to:
“(E) Bone spicule pigmentation with greater density at nasal side was noted on ultra-widefield (UWF) colour photograph and (F) a typical macular hyperautofluorescence ring (HAR) was observed on UWF auto-fluorescence (AF), (G) short-wavelength AF (SWAF) and (H) near-infrared AF (NIAF). However, the borders of HAR could be delineated only on NIAF. (I) Spectral-domain optical coherence tomography (SD-OCT) showed shortening of the ellipsoid zone (EZ).”
- Line 308 in the section “phenotypic patters”. “(patients 1477)” please remove the “s”
- It has been previously determined that variable levels of expression of CNOT3, a trans-acting epistatic factor is genetically linked to PRPF31 and regulates expression of PRPF31. An intronic variant in CNOT3 determines its level of expression and thus how efficiently PRPF31 expression is downregulated. The alleles of CNOT3 inherited determine the expression of non-mutant PRPF31 and thus whether a person will be affected by the disease (Venturini et al., 2012; Rose et al., 2014). Did you test your patients for the CNOT3 intronic variant? Please include a discussion of CNOT3 regulation in reference to your findings.
Author Response
Response to Reviewer 1 Comments
Point 1: Please give the stimulus intensity range of the MAIA and HFA.
Response 1: Thank you for your comment. We have included stimulus intensity ranges in the revised version. Methods, Page 3, paragraph 1, lines 100–111 has been revised to:
“Standard automated perimetry (24-2 pattern, 54 test loci, III-white stimulus, stimulus intensity range 0–40dB) was performed using the Humphrey Field Analyzer (HFA-II 750, Carl Zeiss Meditec GmbH, Germany). HFA 24-2 mean deviation (MD), which is calculated based on the normative values provided on the device, was recorded. Baseline and follow-up microperimetry (Macular Integrity Assessment, MAIA, Centervue, Padova, Italy) with a stimulus intensity range of 0–35dB were performed. The large 10-2 (68 test loci) and the small radial (37 test loci, 37R) grid patterns were used to map the retinal sensitivity profile within the macular (central 20° field) and foveal (central 6° field) regions, respectively (Supplementary Figure S1). Retinal sensitivity was defined as normal (≥ 26dB), scotoma (0–25dB) or dense scotoma (< 0dB), according to the manufacturers’ recommendation.”
Point 2: In the legend of Figure 1, it reads “….present study are marked with WARD study ID.” What is WARD? Please define in the manuscript.
Response 2: Thank you for your comment. Methods, page 2, paragraph 5, line 83, has been revised to:
“… Western Australian Retinal Degeneration (WARD) study….”
Point 3: Please indicate how “normal” is assigned to perimetry results. Did you establish a normative database or did you use values provided on the device?
Response 3: Normal values for the standard automated perimetry are incorporated in the device and used as reference for calculating deviation plots and mean deviation (MD). For the MAIA, we used the classification system recommended by the manufacturer. We have revised the Methods section as follows to include these statements. Page 3, paragraph 1, lines 100–111 has been revised to:
“Standard automated perimetry (24-2 pattern, 54 test loci, III-white stimulus, stimulus intensity range 0–40dB) was performed using the Humphrey Field Analyzer (HFA-II 750, Carl Zeiss Meditec GmbH, Germany). HFA 24-2 mean deviation (MD), which is calculated based on the normative values provided on the device, was recorded. Baseline and follow-up microperimetry (Macular Integrity Assessment, MAIA, Centervue, Padova, Italy) with a stimulus intensity range of 0–35dB were performed. The large 10-2 (68 test loci) and the small radial (37 test loci, 37R) grid patterns were used to map the retinal sensitivity profile within the macular (central 20° field) and foveal (central 6° field) regions, respectively (Supplementary Figure S1). Retinal sensitivity was defined as normal (≥ 26dB), scotoma (0–25dB) or dense scotoma (< 0dB), according to the manufacturers’ recommendation.”
Point 4: Figure 2C and D have hyper AF optic disks. Is this significant to the phenotype? Is it ONH drusen? If this is a finding, please include in the manuscript to further expand the description of the phenotype associated with PRPF31.
Response 4: Thank you for noting this. We agree that optic disc drusen is a well-known association of retinitis pigmentosa. However, we did not find any evidence of optic disc drusen on other imaging modalities. The increased autofluorescence signal at the optic nerve heads in Figures C and D is related to the use of green autofluorescence and image contrast enhancement for visualizing the details. We have added the following statement to the caption of Figure 2 in page 6, line 228 to clarify this:
“Bilateral increased autofluorescence signal in the optic nerve heads in Panels C and D is due the use of green excitation induced autofluorescence and brightness adjustment to improve contrast.”
Point 5: Figure 3, please describe and indicate A, B, C etc in the text for the given panels. Please tell what the arrows are indicating. The legend is confusing when 2 parentheses are side-by-side. For example, “auto-fluorescence (AF) (F),” should be changed to “auto-fluorescence (AF; F)”. Or, put the (F) before discussing what is observed. So, “Bone spicule pigmentation with greater density at nasal side was noted on ultra-widefield (UWF) colour photograph (E) and a typical macular hyperautofluorescence ring (HAR) was observed on UWF auto-fluorescence (AF) (F), short-wavelength AF (SWAF) (G) and near-infrared AF (NIAF) (H). However, the borders of HAR could be delineated only on NIAF (H). Spectral-domain optical coherence tomography (SD-OCT) showed shortening of the ellipsoid zone (EZ) (I).” would be changed to: “(E) Bone spicule pigmentation with greater density at nasal side was noted on ultra-widefield (UWF) colour photograph and (F) a typical macular hyperautofluorescence ring (HAR) was observed on UWF auto-fluorescence (AF), (G) short-wavelength AF (SWAF) and (H) near-infrared AF (NIAF). However, the borders of HAR could be delineated only on NIAF. (I) Spectral-domain optical coherence tomography (SD-OCT) showed shortening of the ellipsoid zone (EZ).”
Response 5: Thank you for your suggestion. Panels of Figure 3 have been added to the relevant section of the manuscript (page 7, paragraph 1, lines 231–235). The caption of Figure 3 (page 7) has been revised according to the reviewer’s comment and arrows have been explained as follows:
“Arrows show the position and direction of the B-scans. Arrowheads indicate borders of preserved EZ.”
Point 6: Line 308 in the section “phenotypic patters”. “(patients 1477)” please remove the “s”
Response 6: Thank you. This error has been corrected.
Point 7: It has been previously determined that variable levels of expression of CNOT3, a trans-acting epistatic factor is genetically linked to PRPF31 and regulates expression of PRPF31. An intronic variant in CNOT3 determines its level of expression and thus how efficiently PRPF31 expression is downregulated. The alleles of CNOT3 inherited determine the expression of non-mutant PRPF31 and thus whether a person will be affected by the disease (Venturini et al., 2012; Rose et al., 2014). Did you test your patients for the CNOT3 intronic variant? Please include a discussion of CNOT3 regulation in reference to your findings.
Response 7: Thank you for your comment. Exploration of the role of CNOT3 expression on PRPF31 level and disease phenotype is beyond the scope of the current manuscript. However, the CNOT3 rs4806718 SNP was genotyped in all patients and is the subject of an upcoming paper. We have added the following sentences to the Discussion, page 13, paragraph 2, lines 405–410:
“Multiple cis- and trans-acting factors such as CNOT3 [48,49] and minisatellite repeat (MSR) elements [50] regulate the expression of PRPF31. CNOT3 binds to PRPF31 core promoter and downregulates its expression, whilst inhibition of CNOT3 upregulates PRPF31 expression [48,49]. These findings suggest that epistatic factors have a role in PRPF31 expression by the wild-type allele and should be considered in future investigations.”

Reviewer 2 Report
In this work, the authors have demonstrated that the retinopathy (RP11) phenotype may be related to the wild-type PRPF31 allele rather than the type of mutation.
I am not an expert in this pathology, but the work is interesting, and the methods are sounds. The authors also reported a limitations section. My minor comment is that, at the beginning of the discussion section, the authors could highlight better the added value of their work..
Author Response
Response to Reviewer 2 Comments
Point 1: My minor comment is that, at the beginning of the discussion section, the authors could highlight better the added value of their work.
Response 1: Thank you for your suggestion. Discussion, page 12, paragraph 1, lines 333–337 has been revised to:
“This study characterises RP11 patterns of disease using multimodal retinal imaging in 7 unrelated adRP families with different classes of PRPF31 mutations and suggests that RP11 phenotype can be classified into early- or late-onset and slow or rapid progression. Such classification may be useful in future genotype-phenotype correlation and natural history studies, as well as RP11 clinical trials.”

Reviewer 3 Report
Reviewer’s Comments:
The manuscript " Clinical evidence of implication of wild-type PRPF31 allele in phenotypic expression of RP11" by Roshandel et al suggests that the RP11 phenotype may be related to the wild-type PRPF31 allele rather than the type of mutation.
Comments:
- Why was only the right eye progression data presented?
- Please add a brief paragraph on future directions at the end of the conclusions section.
- Please be consistent with the style of references.
Author Response
Response to Reviewer 3 Comments
Point 1: Why was only the right eye progression data presented?
Response 2: Previous reports have shown that RP11 affects both eyes symmetrically. Our cross-sectional findings support a symmetrical disease as it is presented in the Supplementary Tables S2 and S3. Hence, data for one eye (usually the right eye or the better eye) reflect the clinical course of disease progression and results of the fellow eye does not add to the results.
Point 2: Please add a brief paragraph on future directions at the end of the conclusions section.
Response 2: Thank you for your suggestion. The following statement has been added to the end of Conclusions, page 14, paragraph 2, lines 448–450:
“We recommend further investigation of tissue PRPF31 levels using patient-derived retinal tissue [51] that may provide additional clues towards a better understanding of these relationships and finding potential therapeutic targets.”
Point 3: Please be consistent with the style of references.
Response 3: References have been checked thoroughly for consistency.
